# FAT/CD36 Participation in Human Skeletal Muscle Lipid Metabolism: A Systematic Review

**DOI:** 10.3390/jcm12010318

**Published:** 2022-12-31

**Authors:** Arnulfo Ramos-Jiménez, Ruth A. Zavala-Lira, Verónica Moreno-Brito, Everardo González-Rodríguez

**Affiliations:** 1Instituto de Ciencias Biomédicas, Universidad Autónoma de Ciudad Juárez, Anillo Envolvente del PRONAF y Estocolmo S/N, Ciudad Juárez 32310, Chihuahua, Mexico; 2Facultad de Medicina, Circuito Universitario Campus II, Universidad Autónoma de Chihuahua, Chihuahua 31124, Chihuahua, Mexico

**Keywords:** fat oxidation, fatty acid-binding protein, scavenger receptor type B2, adenosine monophosphate activating protein, muscle contraction, nutrients, mitochondria, sarcolemmal

## Abstract

Fatty acid translocase/cluster of differentiation 36 (FAT/CD36) is a multifunctional membrane protein activated by a high-fat diet, physical exercise, fatty acids (FAs), leptin, and insulin. The principal function of FAT/CD36 is to facilitate the transport of long-chain fatty acids through cell membranes such as myocytes, adipocytes, heart, and liver. Under high-energy expenditure, the different isoforms of FAT/CD36 in the plasma membrane and mitochondria bind to the mobilization and oxidation of FAs. Furthermore, FAT/CD36 is released in its soluble form and becomes a marker of metabolic dysfunction. Studies with healthy animals and humans show that physical exercise and a high-lipid diet increase FAT/CD36 expression and caloric expenditure. However, several aspects such as obesity, diabetes, Single Nucleotide polymorphisms (SNPs), and oxidative stress affect the normal FAs metabolism and function of FAT/CD36, inducing metabolic disease. Through a comprehensive systematic review of primary studies, this work aimed to document molecular mechanisms related to FAT/CD36 in FAs oxidation and trafficking in skeletal muscle under basal conditions, physical exercise, and diet in healthy individuals.

## 1. Introduction

Obesity (body mass index ≥30 kg/m^2^) is well recognized as a global epidemic; it is an independent risk factor for stroke, hypertension, cardiovascular disease, cardiovascular disease mortality, atrial fibrillation, and sudden cardiac death [1,2]. Losing fat mass, especially in the abdominal zone or visceral adipose tissue, helps to reduce hypertension and improve metabolic syndrome, associated systemic inflammation, and endothelial dysfunction [1,2]. Obesity occurs when there is a chronically positive caloric balance. This imbalance seems simple to solve but does not, because obesity is a complex multifactorial phenomenon challenging to observe and measure in all dimensions [3,4]. Several individuals with obesity have metabolic dysregulation and adiposopathology (fat stores in body locations where fat is not physiologically stored, such as the liver, pancreas, heart, and skeletal muscle), also named sick fat [5]. Many non-surgical treatments designed to reduce body weight and fat propose low-energy diets, increase energy expenditure, and high fatty acids (FAs) oxidation, such as hypocaloric/ketogenic diets, low and moderate-intensity physical exercise, and high-intensity interval training (HIIT) [6,7,8]. However, diet and physical exercise treatments are hard to apply correctly due to physical and psychological differences, and the gene expression variability between individuals and populations [3,9], hence the importance of continuing to study the mechanisms that help to understand the obesity problem.

Of the genetic factors and mechanism of intermediary metabolism associated with obesity, there are those involved in the capture, transport, and oxidation of lipids. The Fatty Acid Translocase/Cluster of Differentiation 36 (FAT/CD36) is responsible for the capture and translocation of FAs across the cell membrane and the outer mitochondrial membrane [10,11]. The Fatty Acid Binding Proteins (FABPs) along with the Fatty Acid Transport Proteins (FATPs) are responsible for the FAs trafficking in the aqueous environment of cells [11]. Variations in the transcription and translation of these proteins have been associated with obesity, metabolic syndrome, and neurodegenerative disorders [12,13], disorders that could be prevented by increasing mitochondrial oxidation and caloric expenditure, especially in skeletal muscle. In this sense, skeletal muscle contributes mostly to the total caloric expenditure because of its size and function. [14]. However, it has been widely reported that skeletal muscle metabolism is altered or works differently in subjects with obesity and diabetes compared to lean and healthy people [15]. Baker et al. report that plasma short-chain acylcarnitine species, skeletal muscle medium-chain acylcarnitine and FAT/CD36 increased from pre- to post-high fat diet (HFD) in obese but not lean subjects [3]. The above mean that although fatty acids (FAs) increase the expression of FAT/CD36, they could also impair β-oxidation. Pepino et al. report that the FAT/CD36 protein is regulated differently in each tissue to adjust to particular energy needs [16]. For that, several literature reviews have focused the implications of FAT/CD36 on FAs uptake, translocation and oxidation and its effect on energy homeostasis and obesity [17,18], supporting the idea of reduced fatty acid oxidation in the skeletal muscle of obese vs. lean individuals [18]. Given the non-conclusive evidence observed, it was proposed to investigate, through a comprehensive systematic review of primary studies, the relationship between human muscle skeletal FAT/CD36 expression, synthesis, and content and how diet and exercise modify them. Therefore, this work presents the best-known mechanisms related to FAs oxidation and FAT/CD36 in human skeletal muscle, and their relationship with diet and physical exercise. The hypothesis is that the skeletal muscle FAT/CD36 protein is regulated by nutritional status, diet, and physical exercise. The main contribution of this work is to elucidate the implications of FAT/CD36 on metabolic adaptations induced by nutritional status, physic al exercise, and high fat diets in human skeletal muscle.

## 2. Materials and Methods

### 2.1. Literature Search Strategy

This review was conducted in accordance with the Preferred Reporting Items for Systematic Reviews and Meta-analyses statement guidelines [19]. An exhaustive systematic search of electronic databases was conducted up to 9 October 2022, with previous searches on 10 May and 5 September 2022, including MEDLINE and BIVIR as institutional database integrator [20]. Although the research protocol was not previously published, all papers that met the inclusion and exclusion criteria were reviewed and evaluated. The search strategy comprised the following key phrases: FAT/CD36, SR-B2, diet, nutrition, nutrient, exercise “energy expenditure” “energy metabolism” to identify relevant trials. The boleans and syntax used was: (FAT/CD36 OR CD36 OR SR-B2) AND (diet OR nutrition OR nutrient) AND (exercise OR “energy expenditure” OR “energy metabolism”). The search strategy was limited to human subjects where the option was available. The reference lists of the incl uded studies were also examined for any new references not found during the initial electronic search. Two reviewers independently appraised papers (R.A.Z.L. and A.R.J.); a third reviewer (V.M.V.) was consulted to resolve disputes.

The Strengthening the Reporting of Observational Studies in Epidemiology (STROBE) statement [21] was applied to analyze the strengths and weaknesses of the included studies (Appendix A).

### 2.2. Inclusion and Exclusion Criteria

#### 2.2.1. Type of Study

The search included primary studies involving the uptake, transport, and oxidation of fatty acids by FAT/CD36 protein in human skeletal muscle, with a particular interest in dietary treatments and physical exercise. Longitudinal, cross-sectional, and crossover designs were included, written in English, Spanish or Portuguese. No restrictions were placed on the publication date. Animal studies were excluded from analysis.

#### 2.2.2. Type of Participants

Human subjects.

#### 2.2.3. Type of Interventions

Acute and long diet and physical exercise trials.

### 2.3. Outcomes

The primary outcomes were muscle-skeletal changes in fatty acids metabolism and FAT/CD36 produced by diet and physical exercise. Secondary outcomes were associations between FAT/CD36 expression and nutritional status.

### 2.4. Data Synthesis

Two reviewers (R.A.Z.L. and A.R.J.) extracted data in duplicate and cross-checked results. Zotero (v 6.0.15 free version for Mac, Corporation for Digital Scholarship, Fairfax, VA, USA) was used as a Reference Manager. Outcomes (author and year of publication, study design and population, the purpose of the study, type of intervention, and main results on the expression and activity of FAT/CD36 protein) were extracted and archived in a database for analysis.

## 3. Results

### 3.1. Descriptive Analysis

As shown in Figure 1, using the keywords described in the methods and the chosen metasearch engines, we found a total of 1970 manuscripts; 342 were repeated manuscripts. According to the inclusion criteria and the reading of titles and abstracts, 1469 were ineligible, leaving 159 for full-text reading. Finally, 139 manuscripts were excluded: seventy-nine being animal studies, fourteen for not presenting results on FAT/CD36, forty-five for being review articles, and one for containing the same population and data in two different journals.

Of the twenty manuscripts eligible for the synthesis and analysis, ten were cross-sectional, eight were longitudinal, and two were crossover studies (Table 1). In total, three-hundred and thirteen participants were included (41.5% women), three-hundred and three young adults, and seven older adults. Regarding their nutritional status, 208 were lean subjects, 31 were overweight, and 55 were obese. Regarding the type of study, 117 participated in dietary studies, 172 in physical exercise studies, and 23 in both classes. Regarding their physical activity level, 132 participants were sedentary, 121 were moderately active, and 60 were athletes.

### 3.2. Interventions Analysis

Only three authors mention having performed a randomization of the participants, one of them blind selection (Figure 2 and Figure 3). Moreover, according to the STROBE criteria, to qualify the strength and studies’ weaknesses, all of them obtained a score of 19/22. The three main weaknesses were:the lack of information on the eligibility criteria;not mentioning the measures taken to avoid or reduce possible bias, and;not explaining how to determine the sample size and randomization.

However, due to the nature of most studies (basic science, and assessing biological mechanisms), we do not consider that these three criteria biased the results, much less that it detracts from the studies quality.

Regarding diet, cross-sectional (five studies with 73 individuals) and crossover (one study with 14 individuals) design studies show that a high-fat meal increases the gene expression and abundance of the FAT/CD36 protein in skeletal muscle in lean and obese individuals. High glycemic index (HGI) meal decreases FAT/CD36 mRNA and protein levels (one study with 8 individuals). Regarding physical exercise, short sessions of endurance training and HIIT type increased the expression and amount of FAT/CD36 protein in skeletal muscle (eight studies with 116 individuals). The expression and quantity of the FAT/CD36 protein were associated with the activation of genes, increases in the synthesis and activation of FAs transport proteins, oxidation of fatty acids, hydrolysis of lipids and intermediates; among them, Uncoupled protein 3 (UCP3) gene, FABP4, Long-chain Fatty Acid Transport Protein 1 and 4 (FATP 1, FATP 4), Carnitine Palmitoyl Transferase 1 (CPT 1), β-hydroxy acyl-CoA dehydrogenase (β-HAD), muscle lipoprotein lipase (mLPL), Peroxisome proliferator-activated receptors (PPAR), nuclear-encoded protein peroxisome proliferator-activated receptor gamma coactivator (PGC) 1, Pyruvate Dehydrogenase Kinase (PDK) 4, Citrate Synthase (CS), 5-AMP-activated protein kinase (AMPK), Extracellular Signaling Receptor Kinase (ERK), and Protein Kinase C (PKC).

## 4. Discussion

### 4.1. Characterization and Localization of FAT/CD36

Several FAs transport proteins work together, including FABPs, the FATP family, and FAT/CD36. They all facilitate the uptake and transport of FAs across the plasma and mitochondrial membranes and their subsequent oxidation, where FAT/CD36 is the main LCFAs transport protein in sarcolemma and mitochondrial membranes [10,38]. The multifunctional human protein FAT/CD36 (Figure 4) is an 88 kDa transmembrane glycoprotein, a member of a superfamily of Scavenger Receptor Proteins class B (SR-B) [39]; the gene is located on chromosome 7 (7q11.2). FAT/CD36/SR-B2 is encoded by 15 exons and is formed by a single chain of 472 amino acids [40,41]. It has two transmembrane areas, two short intracellular domains, a large extracellular loop and a hairpin membrane topology with two transmembrane regions [42]. Both NH2 and COOH ends are short segments anchored in the cell cytoplasm [42]. Moreover, it has two phosphorylation sites, three external disulfide bridges, and four palmitoylation sites, two at the NH2 and two at COOH terminals. The last part of the protein is the COOH terminal domain of FAT/CD36, which contains two ubiquitination sites [41].

FAT/CD36 is expressed in various tissues, such as the liver, heart, skeletal muscle, adipose tissue, endothelium, and blood cells [42,43]. Schenk and Horowitz reported in 2006 that FAT/CD36 and CPT1 proteins coimmunoprecipitate in skeletal muscle [33] so both are found in the outer mitochondrial membrane. Later in 2011, Smith et al. demonstrate that FAT/CD36 is positioned on the outer mitochondrial membrane, upstream of long-chain acyl-CoA synthetase activating FAs [44]. In 2022, Zeng et al. confirmed that FAT/CD36 is positioned on the upstream of long-chain acyl-CoA synthetase, thereby contributing to the FAs activation and regulation of mitochondrial Fas transport [45]. Nevertheless, FAT/CD36 is not in mitochondrial contact sites nor directly interacting with CPT-1, but with the FAs in order to activate them and, in turn, increasing the maximal transport (Vmax) of long-chain fatty acid (LCFAs) [46,47]. Its translocation to the mitochondria is like the sarcolemma (Figure 5b). At rest, FAT/CD36, similarly to other proteins (GLUT4), are generally found intracellularly in lipid rafts (VAMP2, a vesicle-associated membrane protein isoform) [42,43], which translocate to their sites of action upon stimulus. During an acute high-fat diet, physical exercise, and muscle contraction, FAT/CD36 is translocated from intracellular sites to the plasma and mitochondrial membrane, induced by several molecules such as insulin, Ca^2+^, AMPK, ERK, and Protein Kinase C (PKC) [48,49,50] (Figure 5). Later, under new resting conditions, FAT/CD36 interacts with caveolins in the sarcolemma for its invagination and recovery in plasma.

### 4.2. Regulation and Functions of FAT/CD36

In a murine model of atherosclerosis, there are formation of foam cells, atherosclerotic plaques, mitochondrial reactive oxygen species high levels, decreased fatty acid oxidation, and LCFA accumulation; all these phenomena are correlated with FAT/CD36 expression. [51,52]. However, in non-pathological conditions, FAT/CD36 (Figure 5) is essential in the homeostasis and trafficking of LCFAs from the interstitial liquid into cells for subsequent oxidation and ATP production [43]. FAT/CD36 expression, integration into the plasma membrane, and translocation through the cell plasma are highly stimulated by several mechanisms: among them N-linked glycosylation [53], AMPK phosphorylation [54], palmitoylation [45,55], leptin [54], insulin [56,57], high-fat diet [48], LCFAs availability in tissue [10,58], physical exercise [9,36], and muscle contraction [42,56,58,59]

FAT/CD36 function in a wide range of processes not always related to FAs uptake, translocation, and oxidation, as apoptosis, angiogenesis, phagocytosis, thrombosis, inflammation, and atherosclerosis [60]. In 2022, Glats et al. noted that LCFAs transport and oxidation in skeletal muscle are upregulated by the expression, protein synthesis and translocation of FAT/CD36 [10]. Moreover, FAs upregulate the expression and synthesis of FAT/CD36 by the activation of transcription factors PPARs (the preferred ligand of FAs) [61]. With this feedback regulation, the FAT/CD36 would cover most energy demands from lipid sources, especially in metabolically highly active tissues such as skeletal muscle. A close relationship has been observed between AMPK activation, FAT/CD36 translocation to the plasmatic membrane, and the subsequent increase in the capture and oxidation of FAs [62]. However, the mechanisms of this association are unknown. In 2015, Monaco et al. found that FAT/CD36 accumulation and translocation to the plasma membrane is independent of AMPK, but high muscle-contraction accelerates it [63]. In 2017, Momken et al. also found that leptin activate AMPK and induce the membranal plasma translocation of FAT/CD36 increasing FAs uptake and FAs oxidation in mice isolated extensor digitorum longus muscle [54]. However, Xu et al. [64], studying FAs transmembrane movement in HEK293 cells found that FAT/CD36 accelerates the uptake and rapid esterification of FAs without necessary catalyzing the translocation of FAs across the sarcolemmal. That is, FAs transport across the sarcolemmal is mainly through diffusion, a process that does not usually require FAT/CD36. Concluding, FAT/CD36 increase the FAs uptake without necessary cause the translocation of FAs across the plasma membrane. These mechanisms are still under study.

### 4.3. Gene Expression and Molecular Heterogeneity of FAT/CD36

Defective FAT/CD36 protein has been related to several health problems as obesity, intestinal fat malabsorption, kidney disease, lipotoxic cardiomyopathy, impaired fatty acid metabolism, glucose intolerance, atherosclerosis, arterial hypertension, diabetes, cardiomyopathy, Alzheimer’s disease, and multiple sclerosis [41,42,65]. Moreover, various FAT/CD36 post-translational modifications, such as glycosylation, acetylation, phosphorylation, palmitoylation, and ubiquitination, impact its function [66]. Some modifications are related to pathologies’ appearance (Table 2). In this sense, there are several Single Nucleotide Polymorphisms (SNPs) at the FAT/CD36 gene related to modifications in protein expression levels, translation, and pathologies of metabolic origin [42]. Furthermore, oxidative stress and a prothrombotic phenotype perturb FAT/CD36 function and induce metabolic diseases as obesity, insulin resistance, diabetes, atherothrombotic disease, chronic kidney disease, neurodegenerative disorders and multiple sclerosis [67,68]. Mutations of the FAT/CD36 gene increase the accumulation of LCFAs in the human heart [69]. Concerning physical activity, Jayewardene et al. report a potential pleiotropic influence of FAT/CD36 SNPs, where TT SNP genotypes at rs1527479 and wild-type GG genotypes at rs1984112 were associated with significantly greater whole-body fat oxidation during submaximal exercise [70]. Koonen et al. mention that plasma soluble FAT/CD36 can be used as a metabolic marker for FAs metabolic dysfunction [71].

### 4.4. Effect of Physical Exercise and Nutrients on FAT/CD36 Regulation and Energy Expenditure

The other purpose of this systematic review is to collect information regarding the FAT/CD36 protein and its interaction with factors such as physical activity and nutritional issues.

#### 4.4.1. Physical Exercise

Although it does not apply in all circumstances, this work defines physical exercise as any bodily movement carried out consciously to improve health or fitness. Aerobic exercise, also called endurance, can be carried out for long periods at low and moderate intensity (heart rate < 80%), using oxidative metabolic pathways to produce ATP. It is well-established that athletes and women use more fat than carbohydrates at any exercise intensity than sedentary people and men as an energy source; however, these differences are in discussion [72,73]. Regarding this work, the study of FAT/CD36 protein during physical exercise has increased over the last twenty years. On this topic, two thematic reviews have recently been published, one on the FAT/CD36 functions in lipid metabolism and signaling [10] and the other on regulating fat metabolism during aerobic exercise [74]. The first one highlights the importance of FAT/CD36 in cellular lipid homeostasis and its relevance as a target for metabolic modulation therapy in disease. The second provides a fascinating narrative of the FAs transport from the triacylglycerols lipolysis in muscle and adipocyte, to the bloodstream, and finally their entry into cells and mitochondria until its oxidation in the mitochondria matrix. In this sense Pelsers et al. [75], and Holloway [47] report that physical exercise is the primary source of increased oxidation of FAs, and a sedentary lifestyle is the leading cause of the imbalance between the entry of FAs and their oxidation in the mitochondria, increasing the accumulation of TAG in skeletal muscle and producing insulin resistance. LCFAs from the bloodstream are the primary source of skeletal muscle energy during low- and moderate-intensity exercise, especially for the type-1 fibers [75]. As detailed above, LCFAs enter cells via passive diffusion along the concentration gradient across the sarcolemma or facilitated membrane proteins. This facilitated pathway is highly regulated, either for the availability of LCFAs or various complex processes not yet fully understood. The facilitated pathway during the exercise is the top way to meet the energy demand. This section discusses the effect of physical exercise on FAs oxidation, energy expenditure, and the role of FAT/CD36 in skeletal muscle (Table 2).

In a cross-sectional study in 2007, Yanai et al. reported that participants with FAT/CD36 deficiency showed significantly lower ventilatory threshold (VT) than normal participants after 15 min of the incremental exercise test. Moreover, a high correlation (R = 0.785) was observed between VT and decreased plasma FAs [76]. Holloway et al. through the Western Blotting technique in mitochondria isolated by centrifugation gradients, report a gradual increase in mitochondrial FAT/CD36 protein content (30–60%) and LCFA transport and oxidation (radiolabel [C^14^] palmitate) during acute endurance exercise (120 min at 60% of VO_2_ peak) in skeletal muscle [11]. Recently in 2022, Maunder et al. found that FAT/CD36 abundance in skeletal muscle correlated with peak fat oxidation (R = 0.68) and CS activity (R = 0.84) [31]. In 2008, Perry et al., after 18 h of HIIT, reported an increase in the content and activity of oxidative and glycolytic proteins, improving skeletal muscle capacities to oxidize fat and carbohydrate in previously untrained individuals: increased Cytochrome C oxidase IV (Cox-IV, 18%), CS (26%), β-hydroxyacyl-CoA dehydrogenase (β-HAD, 29%), aspartate-amino transferase (AST, 26%), pyruvate dehydrogenase (PDH; 21%), FAT/CD36, FABP4, Glucose Transport (GLUT) 4, and monocarboxylate transporter (MCT) 1 and 4 (14–30%) [32]. Moreover Talanian et al. (2007, 2010) note that six weeks of HIIT (ten 4-min cycling bouts at 90% VO_2_ peak separated by 2 min of rest) increased FAT/CD36 protein at whole muscle (10%) and mitochondrial levels (51%) in skeletal muscle of untrained females [66]. In 2020, Warren et al. reported that after 8–16 weeks of endurance exercise training (3 days per week, 20–40 min, 67–80% of maximal HR), the FAT/CD36 muscle content and mitochondrial respiratory capacity increased [37]. In 2002, Tunstall et al., after 9-day endurance exercise (60 min cycling per day at 63% of VO_2_ peak; 104 ± 14 W), detected increased gene expression and protein content of FAT/CD36 and CPT I mRNA during 1-h of cycling bout; this also increased the total lipid oxidation by 24% [36]. In summary, moderate-intensity exercise, HIIT, and aerobic type increase the content and enzymatic activity of proteins responsible for the translocation, transport, and oxidation of FAs. However, the above studies have the weakness of not controlling the participants’ diet. The type of diet affects the FAT/CD36 gene transcription and translation and FAT/CD36 protein translocation [24,48] altering the FAs metabolism. These effects will be discussed later. An 8–16-week aerobic exercise training (3 day/wk, 20–40 min, 67–80% of maximal HR) improved FAs oxidation and CD36 mRNA content in skeletal muscle (healthy premenopausal women); the authors conclude that this improvement was primarily due to mitochondrial biogenesis and not intrinsic mitochondrial oxidative enzymes. [37]. However, Warren et al. indirectly measures mitochondrial biogenesis by high-resolution respirometry (Oroboros Oxygraph O2K). The above observations must be reinforced by analyzing a larger group of proteins related to the signaling pathways of mitochondrial biogenesis, such as nuclear respiratory factor-1 (NRF-1), NRF-2, and estrogen-related receptor-α (ERR-α), and by the increase in expression of TFAM, the final effector of mtDNA transcription and replication. In addition, confocal microscopy could also more accurately quantify mitochondrial populations in muscle tissue.

There are no differences in LCFA oxidation and FAT/CD36 protein mitochondrial content between lean and obese people, and it seems that obesity does not alter the ability of skeletal muscle mitochondria to oxidize FAs at rest and during exercise. However, the systemic inflammation productid by obesity disturbs the immune system, activating the macrophages and altering de FAT/CD36 signaling pathways [77]. Holloway et al. (2007) and Holloway, Bonen, and Spriet (2009) report that there is a lower enzymatic activity of β-HAD, CS, and Cox IV, content and, therefore, lower palmitate oxidation in skeletal muscle of obese vs. lean women, but similar FAT/CD36 protein content; however, FAT/CD36 protein content was positively correlated with mitochondrial fatty acid oxidation (r = 0.67, *p* < 0.05) [28,47]. In 2012, Greene et al. report that after twelve weeks of endurance treadmill exercise (3 sessions/wk, progressing to 500 kcal/session) there is an increase in FAT/CD36, CPT I, Cox IV, LPL, PPAR-δ and PGC-1α protein content in overweight and obese men and women [26]. Therefor physical exercise is necessary for a healthy oxidative metabolism because it acutely and chronically increases lipid oxidation, stimulating various genes that control mitochondrial function and oxidative metabolism.

Frandsen et al. recently reported that the response of energy metabolism to one long-duration aerobic exercise is different between old and younger men [9]. During a bicycle race (36 and 40 h), the energy balance decreased more in young than older men. After the race the VO_2_ max and heart rate, but not FAs oxidation, were only reduced in old men. Furthermore, maximal fat oxidation (MFO), FAs oxidation, and plasma FAs decrease more in young men. After the race, the FAT/CD36, FATP-4 proteins, and the intramuscular triacylglycerols increase significantly in older men but not in the younger [9]. In other words, there is an apparent central adaptation to long-duration and extreme exercise in adults vs. young, observed by a decrease in heart rate at the end of the race in adults. However, physical exhaustion was higher in adults, observed by the decrease in VO_2_ max. Both groups improved insulin and leptin sensitivity, observed by a decrease in plasma concentrations of both. However, the concentrations of FAs in plasma and its maximum oxidation decreased. FAT/CD36 and FATP4 proteins increased in adults but not in young people. That means that age modifies adaptations to physical exercise. Clinically, this may suggest that treatments to increase fat energy expenditure are very specific between different populations.

Regarding the sex differences, Kiens et al. found that FAT/CD36 mRNA and FAT/CD36 protein in skeletal muscle increase by acute exercise, and is higher in women than in men, irrespective of training status [30]. The differences in muscle fiber-type specific contents could partly explain the sex differences in FAT/CD36 expression and FAs oxidation. A well-recognized sex difference (sexual dysmorphism) in skeletal muscle morphology and metabolism exists, as skeletal muscle tissue tends to contain more type I muscle fibers in females vs. males [73,78]. In contrast, the proportion of type IIA or both IIA and IIX is more remarkable in men [79]. Moreover, the sex-specific differences in skeletal muscle FAT/CD36 protein content could also partly be due to the higher β-oxidation enzymes content and LCFA flux in females vs. males [73,80], promoting the gene expression and synthesis of FAT/CD36. In short, skeletal muscle FAT/CD36 protein content depends on muscle fiber-type composition, sex, and training status. However, based on respiratory exchange ratio (RER) and respiratory quotient (RQ) Roepstorff et al. report no sex difference in the relative contribution of carbohydrates and lipids to the oxidative metabolism across the leg (vastus lateralis) at rest and during submaximal exercise (90 min, 58% of VO_2_ peak) [81].

#### 4.4.2. Diet

Various animal studies show that cellular and body energy metabolism is modified by food and diet, where FAT/CD36 protein plays a significant role in this process. Moreover, differences in macronutrient composition and dietary glycemic index (GI) are sufficient to alter fat metabolism. This section discusses the literature on the relationship between FAT/CD36 in human skeletal muscle, meal, and diet.

FAT/CD36 gene transcription, translation and protein content and translocation are sensitively regulated by food and diet. In 2004, Arkinstall et al. found that after two isoenergetic meals [(235 kJ/kg of body mass (BM)], the low-carbohydrate diet (0.7 g/kg BM of CHO, 4.4 g/kg BM of fat, 4 g/kg BM of protein) vs. the high-carbohydrate diet (10 g/kg BM of CHO, 1 g/kg BM of fat, 1.9 g/kg BM of protein) increases the mRNA abundance of FAT/CD36, hormone-sensitive lipase (HSL), β-HAD, and uncoupling binding protein-3 (UCP3) in skeletal muscle [48]. In 2003, Cameron-Smith et al. found that one acute (48–120 h) fat diet (>65% of daily energy) increases the skeletal muscle gene expression of FATP, FAT/CD36, and β-HAD [23]. In 2014, Jordy et al. found an increase in intramuscular triacylglycerol content, HOMA-IR-Index and mRNA content of FAT/CD36 and FABP4 after three days of hypercaloric and high-fatty diet (175% of energy of estimated daily energy intake, 77% from fat) [29]. In 2015, Most et al. found that acute green tea intake (epigallocatechin-3-gallate; 282 mg/day) increases FAT/CD36 expression in adipose tissue, but not muscle lipolysis, whole-body-fat oxidation nor energy expenditure in overweight individuals [82]. In 2014, Gerling et al. reported an increase in resting metabolic rate (5.3%) via an increase in UCP3 long form (11%), without changes in mitochondrial FAT/CD36 and mitochondrial biogenesis after for 12 weeks of Omega-3 supplementation (3.0 g/day) in healthy young men [25].

The association between food intake and caloric expenditure is also due to hormonal changes, induced by food effect on the entire body, e.g., leptin is a central and peripherical glucose and lipid regulator. In the brain, leptin regulates feeding and energy expenditure [83]. Animal and human studies have shown that obesity and high-fat diet induce leptin and insulin resistance [84,85]. Leptin activates AMPK and induces FAT/CD36 protein translocation to the plasma membrane, stimulating the uptake and oxidation of FAs and increasing energy expenditure [54,86]. The uptake and oxidation of FAs by leptin must activate processes other than AMPK phosphorylation since AMPK phosphorylation alone does not increase FAs oxidation [54]. In 2006 and 2009, Schenk et al. reported increases in resting whole-body fat oxidation and skeletal-muscle oxidative capacity after physical exercise; besides, FAT/CD36 content strongly correlated with the increase in whole-body fat oxidation (R^2^ = 0.857) after losing 12% of body weight by exercise and diet (stationary bicycle ergometer for 35–45 min, 70–80% of maximal heart rate, four days/week; 55% carbohydrate, 25% fat, and 20% protein; 500–800 kcal/day below that required to maintain body weight) but not by diet only [86,87]. Cheng et al. mentioned that FAT/CD36 mRNA and protein levels in skeletal muscle decrease below baseline after moderate exercise (60 min, 75% VO_2_max) and a high-glycemic-index diet (70% carbohydrate, glycemic index = 76.6), but not after the only diet in healthy males [24]. Bergouignan et al. (2012) report that the ability to adapt to an acute increase in dietary fat is not impaired in obesity [22]. On the other hand, in 2008 Corpeleijn et al. found no differences in skeletal muscle FAT/CD36 and FABP4 content at the basal level between obese men with and without impaired-glucose tolerance (IGT) [87]. Moreover, skeletal muscle FAT/CD36 protein but not FABP4 increased 1.5-fold after 3 h of insulin-stimulation, similarly between obese men with and without IGT. The above confirms the specific participation of insulin in the intracellular regulation of FAT/CD36 protein concentrations. Moreover, Heilbronn et al. report that subjects with a family history of type 2 diabetes had an impaired ability to increase FAs oxidation in response to a high-fat meal [27]. This inability was related to impaired activation of genes involved in lipid metabolism, including those for PGC1α and FAT/CD36 proteins. In 2010, Corpeleijn et al. found an insignificant −1.9% in fat oxidation in 722 European obese subjects with SNP-78A > C FAT/CD36 (rs2232169) in the fasting state but not after a high-fat diet; 95% energy of fat (60% saturated fat, energy content 50% of estimated resting energy expenditure) [88]. In 2010, Bokor et al. report that four FAT/CD36 SNPs (rs3211908, rs3211867, rs3211883, and rs1527483) were associated with a high risk of obesity in adolescents (OR: 1.73–2.42) [89]. In 2015, Baker et al. found that skeletal muscle mRNA content for genes involved in β-oxidation and mitochondrial biogenesis is low in obese vs. non-obese individuals [3]. Moreover, a high-fat diet increases the concentrations of CS, CPT-1, PDK4, FAT/CD36, UCP-3, Nuclear Receptor Coactivator (NCOA) 1, NCOA2, and transcription factors PPARα, PPARδ, PPARγ, and PGC [3]. Therefore, genetic variability is also associated with FAs oxidation, energy expenditure and body weight changes.

## 5. Conclusions

The synthesis, activity, and translocation of FAT/CD36 protein increase the transport of LCFAs from the blood into the cell and mitochondria for its subsequent oxidation and ATP production; however, its exact transport mechanism has not been elucidated. FAT/CD36 is highly regulated by high-fat diet, physical exercise, LCFAs, Ca^2+^, AMPK phosphorylation, and leptin. Polymorphisms and defects in FAT/CD36 transcription and translation alter normal lipid metabolism, producing obesity, metabolic syndrome, atherosclerosis, arterial hypertension, diabetes, cardiomyopathy, Alzheimer’s disease, and multiple sclerosis. Cross-sectional and Crossover design studies show that a high-fat meal and physical exercise increases the expression and amount of the FAT/CD36 protein in skeletal muscle. Moreover, the expression and quantity of the FAT/CD36 protein in skeletal muscle were associated with improvement in the activation of genes, synthesis and activation of FAs transport proteins, FAs oxidation, and lipids hydrolysis. Some studies show that people with obesity have a lower oxidative capacity, but FAs oxidation and FAT/CD36 content are not affected; studies to confirm this is lacking. Women present higher FAs oxidation and FAT/CD36 protein content than men; studies which confirm this are lacking. In summary, the studies reviewed here indicate that an acute and chronic high-fat diet (50–70% of lipids) and/or endurance exercise (<80% of HR) increase the human skeletal muscle FAT/CD36 mRNA and protein content; its translocation to the cell membrane and mitochondria increases lipid oxidation. Due to the small amount of work, subjects studied, and different methodologies used, the results are not conclusive. Moreover, weight loss treatments should be individualized and consider genetic factors, sex, age, nutritional status, and physical fitness, thus favoring a healthy oxidative metabolism and nutritional status.

## 6. Limitations

Due to the type of analysis in most of the works (cell biology), the sample size is still small, for which a subsequent meta-analysis is necessary to improve the strength of the results.

## Figures and Tables

**Figure 1 jcm-12-00318-f001:**
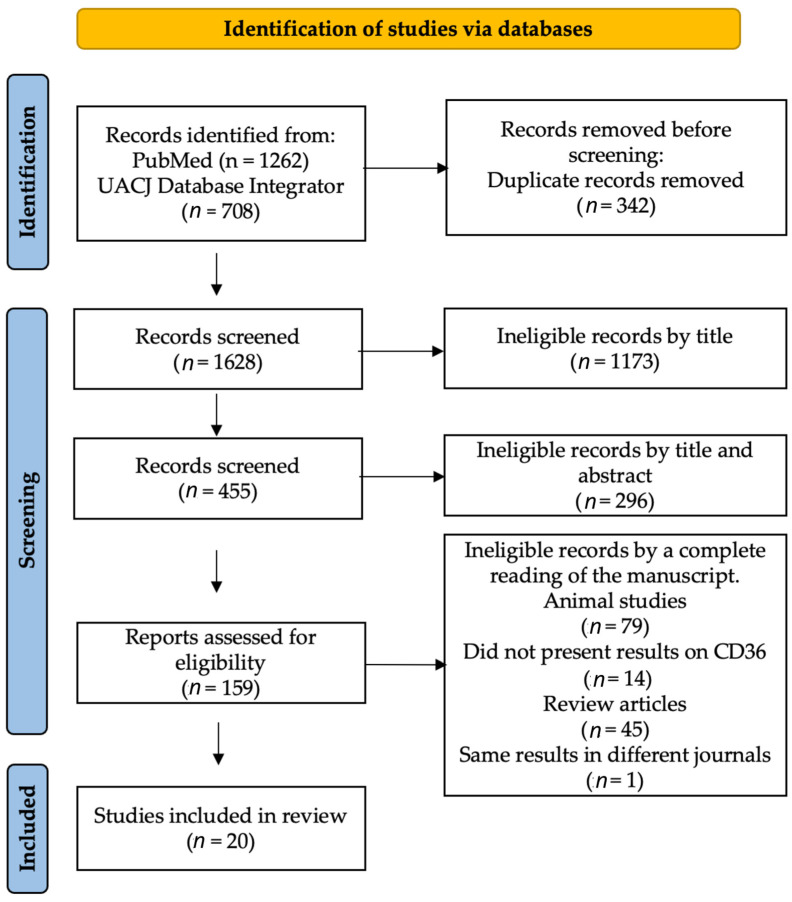
Flow of information through the different phases of a systematic review.

**Figure 2 jcm-12-00318-f002:**
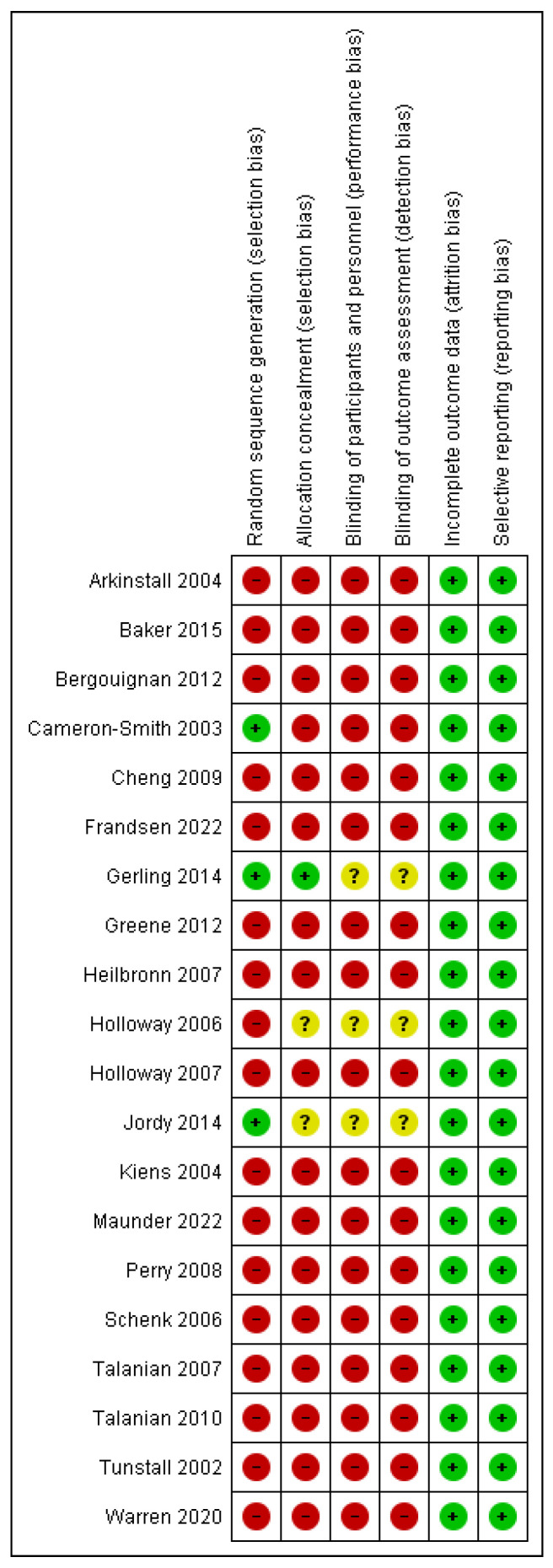
Risk of bias summary: review authors’ judgements about each risk of bias item for each included study [1,3,9,11,22,23,24,25,26,27,28,29,30,31,32,33,34,35,36,37].

**Figure 3 jcm-12-00318-f003:**
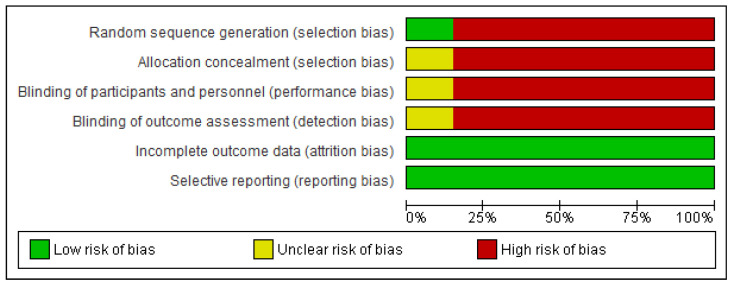
Risk of bias graph: review authors’ judgements about each risk of bias item presented as percentages across all included studies.

**Figure 4 jcm-12-00318-f004:**
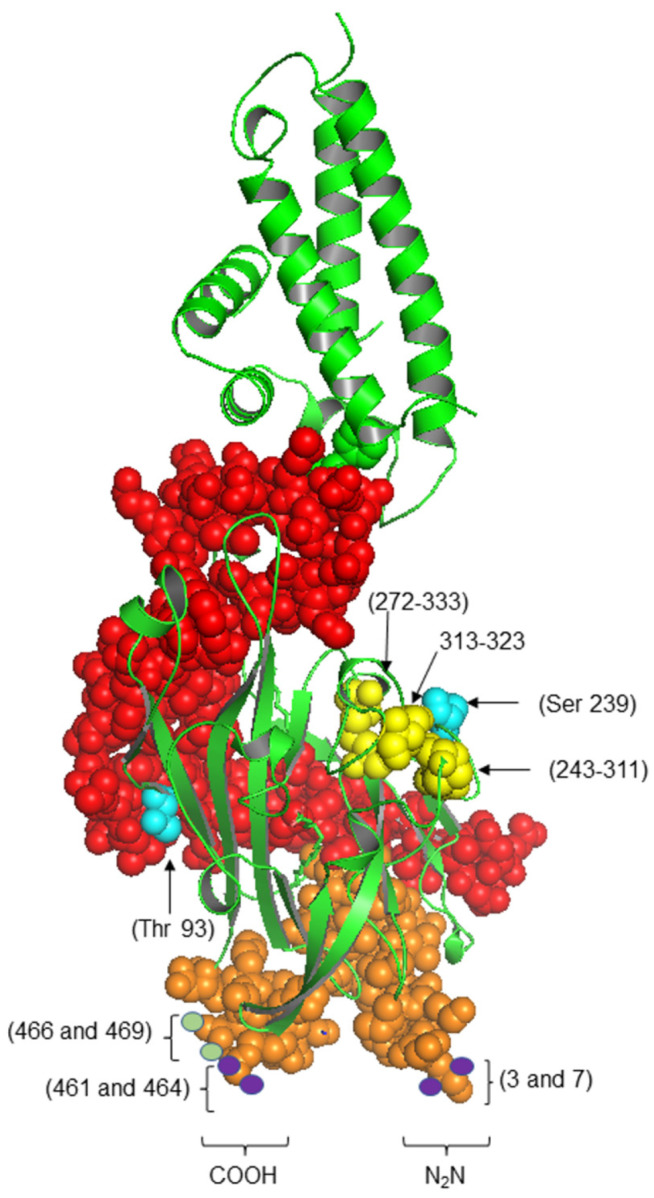
Three-dimensional structure of FAT/CD36 protein. The image represents the sites of the functional relevance of the FAT/CD36 protein. In orange, the intracellular amino and carboxyl-terminal domains, two phosphorylation sites (blue), four palmitoylation sites (purple), three disulfide bond sites (yellow), and two ubiquity sites (green). In red, the hydrophobic pocket is represented through which the fatty acids are presumed to be transported to the outer membrane—a model designed with 5LGD: X-ray Diffraction, 2.07 Å. “The CIDRa domain from MCvar1 PfEMP1 bound to CD36” data deposited and modeling in the SWISS-MODEL database. The graphs were visualized in the PyMOL 4.6.0. program.

**Figure 5 jcm-12-00318-f005:**
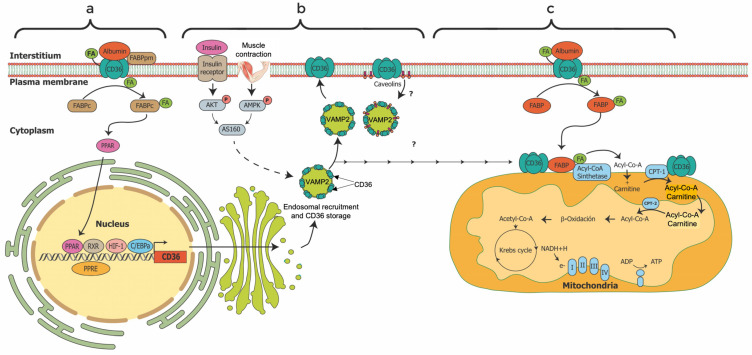
(**a**). Regulation of FAT/CD36 expression. Fatty acids (Fas) internalized into the cytosol are natural ligands of the PPAR transcription factor; this interaction leads to its activation, a condition that induces the transcription of FAT/CD36 through the formation of a heterodimer with “PPAR response element (PPRE)”, the “retinoid X receptor (RXR)”. Additionally, other transcription factors can exert transcriptional control, such as HIF-1 and C/EBPα. (**b**). Diagram illustrating the feedback of FAT/CD36 expression in musculoskeletal tissue. Under resting conditions, FA internalization induces FAT/CD36 synthesis through the molecular mechanism regulated by the transcription factor PPAR. FAT/CD36 is transported across the endomembrane system and ultimately stored in endosomal vesicles. Under conditions of physical activity, muscle contraction, high fat diet, the presence of insulin or leptin, the movement of endosomal vesicles VAMP2, leading to FAT/CD36 to the plasma membrane is induced, increasing their number and consequently the internalization of Fas and their oxidation and production of ATP in the mitochondria. The model also illustrates a possible mechanism that can explain how FAT/CD36 can be transported and located in the mitochondrial membrane through the vesicular transport mechanism, which would confirm the reports regarding the presence of FAT/CD36 in mitochondrial membranes. (**c**). Traditional mechanism of facilitated diffusion of fatty acids (FAs). Long-chain FAs are transported from the extracellular space into the cell for esterification in the cytoplasm or oxidation in the mitochondria. FAT/CD36 actively participates in the transport of FAs, first placing them in its “hydrophobic pocket” (not shown), and later, under a Flip-flop mechanism, the Fas on the outer membrane surface are translocated to the inner membrane. Subsequently, by various phosphorylation mechanisms and activation, the Fas are transported to the surface of the mitochondria by FABPs to be activated by Acetyl-CoA-Synthetase, creating the activated form of FAs (acyl-CoA). Through the action of the enzymes CPT-1 and CPT-2, the acyl-CoA will be transported to the mitochondrial matrix for its final oxidation through the biochemical cycles of B-oxidation, the Krebs cycle, and oxidative phosphorylation. Please consult the list of abbreviations.

**Table 1 jcm-12-00318-t001:** Selected manuscripts according to the inclusion and exclusion criteria.

Author and Year of Publication	Design/Participats	Purpose of the Study	Characteristics of the Intervention	Output
Arkinstall, et al., 2004 [1]	Cross-sectional study.Seven males moderately trained in cycling: age = 35 ± 5 y, BM = 80.3 ± 9.5 kg.	To quantify the acute effect of HCD (77.7%,) and LCD (7.7%) on energy metabolism genes transcription.	Glycogen depletion and HIIT: 2 min 95% and 2 min 55% of VO_2_ peak. LCD (0.7 g/kg of BM of CHO, 4.4 g/kg BM of fat, 4 g/kg BM of protein) or HCD (10 g/kg BM of CHO, 1 g/kg BM of fat, 1.9 g/kg BM of protein).	FAT/CD36 and UCP3 gene transcriptions in skeletal muscle were increased following an acute LCD.
Baker et al., 2015 [3]	Cross-sectional study. Twelve sedentary and healthy caucasian males: age = 19–27 y. 6 lean: BMI ≤ 24.9 kg/m^2^, and 6 with obesity: BMI≤30 kg/m^2^	Investigate the effects of obesity and HFD exposure on fatty acid oxidation and three carboxylic acids cycle intermediates and amino acids in skeletal muscle.	1 and 5 days of HFD: 65% fat, 15% protein, and 25% carbohydrate and comprise 35% of daily energy intake.	HFD increase skeletal muscle FAT/CD36 levels in obese but not in the lean individuals.
Bergouignan et al., 2012 [22]	Cross-sectional study. Nineteen healty participants from 20–45 y: 9 lean men: BMI = 19–25 kg/m^2^, and 10 females with obesity: BMI = 30–40 kg/m^2^	To test that obese non-diabetic humans have an impaired ability to adapt to an HFD.	Isocaloric LFD (20% of energy) and isocaloric HFD (50% of energy).	The expression of skeletal muscle PDK 4, FAT/CD36 and AMPK increased during HFD in both lean and obese individuals during HFD.
Cameron-Smith et al., 2003 [23]	Crossover design.Fourteen well-trained males: age = 26.9 ± 1.7 y, BM = 73.7 ± 1.7 kg.	To determine the effect of HFD on genes expression for FATP and β-oxidation in skeletal muscle.	Five days of HFD (>65% lipids) or HCD (70–75% carbohydrate).	FAT/CD36 and β-HAD gene expression and FAT/CD36 gene abundance were greater after HFD vs. HCD.
Cheng et al., 2009 [24]	Crossover design. Eight healthy males: 22.5 ± 0.3 y, BMI = 22.7 ± 0.5 kg/m^2^.	To determine the effect of meal GI on GLUT4 and FAT/CD36 gene expressions in human skeletal muscle after a single bout of exercise.	60-min cycling exercise at 75% of VO_2_ max, and an isocaloric meal of HGl or LGl with similar proportions of carbohydrate, fat, and protein.	FAT/CD36 mRNA and protein levels were decreased with the HGI vs. LGI meal but not by acute exercise. GLUT4 mRNA was downregulated by both HGI and LGI diets.
Frandsen et al., 2022 [9]	Prospective study.Fourteen well- trained cyclists male.Seven younger: age = 30 ± 5 y, BM~77 kg, and 7 older: age 65 ± 6 y, BM~70 kg.	To examines the physiologic and metabolic impact of repeated prolonged moderate intensity exercise (7–10 h/day for 15 consecutive days at ~63% HR maximal in two cohorts of younger and old cyclists.	3000 km of cycling from Copenhagen, Denmark, to Palermo, Italy over 15 days.	Fifteen days of extreme endurance exercise kept skeletal muscle FAT/CD36 and FABP4 unchanged.
Gerling et al., 2014 [25]	Prospective simple blind randomized study. Thirty healthy, recreationally active males, age~21 y, BMI ~24.6 km/m^2^. 21 intake omega-3 and 9 placebo.	To examine the effects of EPA and DHA supplementation on whole-body RMR and the content of proteins involved in fat metabolism in human skeletal muscle.	12 weeks of 3.0 g/day of EPA and DHA or olive oil.	Twelve weeks of 3.0 g/day of EPA and DHA did no change whole-body fat oxidation, and skeletal muscle mitochondrial content of FAT/CD36, FABP4, FATP 1, and FATP4.
Greene et al., 2012 [26]	Prospective study.Sixteen sedentary overweight and obese partipants: 9 men, age = 41 ± 2 y, BMI = 32.0 ± 2.3 kg/m^2^; 7 women, age 52 ± 2 y BMI = 31.9 ± 1.7 kg/m^2^.	To determine the association of skeletal muscle PPAR content with blood lipids and lipoproteins before and after exercise.	A single session or 12-week program (3 session/day) of land or water treadmill training. 250–500 kcal/ session, 60–85% of VO_2_ max.	A single exercise session, but not exercise training, increases the FAT/CD36, PPAR, PGC-1, and LPL content in skeletal muscle.
Heilbronn et al., 2007 [27]	Cross-sectional study.Five men and twelve women sedentary and nonsmoking. 9 with (FHD-2): age = 46 ± 6 y, BMI= 26.6 ± 5.3 kg/m^2^, and 8 without FHD-2: age= 41 ± 7 y, BMI = 26.7 ± 5.3 kg/m^2^	To examine whole-body glucose and fat oxidation after a prolonged fast and in response to refeeding a single high-fat meal or high-carbohydrate meal meal in both groups.	1000-kcal meal, 76% of high-fat, or 76% of high-carbohydrate.	After a single high-fat meal, the individuals with FHD-2 increase the respiratory quotient and decrease FAT/CD36, CPT1, and PGC1α gene expression in skeletal muscle.
Holloway et al., 2006 [11]	Cross-sectional study.Fiveteen healthy, recreationally active individuals. 10 males and 5 females: age = 22 ± 1 y, BMI = 24 ± 1 kg/m^2^.	To investigate the effects of exercise on CPTI, palmitoyl-CoA and Malonil-CoA kinetics, on the presence and functional significance of FAT/CD36 on skeletal muscle mitochondria.	120 min of cycling at ~60% VO_2_ peak	Whole body fat and palmitate oxidation rates in isolated mitochondria progressively increased during exercise and were correlated (r = 0.78). Skeletal muscle FAT/CD36 protein increased by 63% during exercise and was correlated with mitochondrial palmitate oxidation rates (r = 0.52).
Holloway et al., 2007 [28]	Cross-sectional study.Eighteen nondiabetic women.9 lean: age = 47 ± 3 y, BMI < 27 kg/m^2^; and 9 with obesity: age = 45 ± 3 y, BMI ≥ 30 kg/m^2^.	To examine whether the obesity-related decreases in skeletal muscle lipid oxidation are attributable to (1) a reduction in mitochondrial content and/or (2) an intrinsic defect in mitochondria, and (3) whether there are reductions in the content of mitochondrial FATP	NA	Skeletal muscle FAT/CD36 did not differ in lean and obese individuals but was correlated with mitochondrial fatty acid oxidation (r = 0.67). Obesity did not alter the ability of isolated mitochondria to oxidize palmitate; however, fatty acid oxidation was reduced at the whole muscle level by 28% in the obese.
Jordy et al., 2014 [29]	Cross-sectional study.Eighteen healthy, moderately trained males: age = 24.4 ± 0.7 y, BMI = 23.3 ± 0.5 kg/m^2^	To investigate lipid-induced regulation of FABP4 in human skeletal muscle and the impact on insulin sensitivity.	A hypercaloric (175% of estimated daily energy intake) of HFD (77% from fat) or HCD (80% from carbohydrate) for 3 days.	Three days of a HFD (77% fat) decreased insulin sensitivity but was not associated with a relocation of FAT/CD36 or FABP4 protein to the skeletal muscle sarcolemma. FAT/CD36 and FABP4 mRNA, but not the proteins, were upregulated by increased fatty acid availability.
Kiens et al., 2004 [30]	Cross-sectional study.Forty six healthy and nonsmoking individuals.24 eumenorrheic women: age~26.3 y, body fat~21.4%; and 22 men: age~25.3 y, body fat~13.9%.	To evaluate whether a physical exercise test and sex, influence the FABP4 and skeletal muscle LPL.	An exercise test on a bicycle ergometer at 60% VO_2_ peak for 90 min.	A single 90-min exercise bout increased FAT/CD36 mRNA (25%) and FABP4 mRNA (15%) muscle levels in male and female. FAT/CD36 protein level was 49% higher in women than in men, irrespective of training status. FAT/CD36 mRNA was only higher in untrained women.
Maunder et al., 2022 [31]	Cross-sectional.Seventeen endurance-trained male cyclists and triathletes: age 34 ± 7 y, BMI = 24.5 kg/m^2^.	To assess relationships between PFO measured during fasted incremental cycling, skeletal muscle FAT/CD36 abundance, endurance performance, and fat oxidation rates during prolonged moderate-intensity fed-state exercise.	Incremental cycling exercise test to evaluate the PFO.	FAT/CD36 abundance in skeletal muscle correlated with PFO (R = 0.68) and Citrate Synthase activity (R = 0.84).
Perry et al., 2008 [32]	Prospective study.Three females and 5 males: age = 24 ± 1 y, BMI~22.7 kg/m^2^.	To investigate the ability of 6 wk of HIIT (18 h at 90% of VO_2_ peak) to increase the whole-body CHO and Fat oxidation.	6 weeks of cycle-ergometer HIIT: ~1 h of 10 × 4 min intervals at ~90% of peak oxygen consumption (VO_2_ peak), separated by 2 min rest, 3 day·week^–1^	Eighteen sesions, six weaks of HIIT (3 d/wk) increases fat oxidation (60%) and FAT/CD36, B-HAD, FABP4 (14–30%) protein content.
Schenk and Horowitz, 2006 [33]	Prospective study.Fiveteen abdominally obese premenopausal health women: age~30 y, BMI = 30–40 kg/m^2^, waist circumference >100 cm.	To determine the effect of an endurance exercise training and weight loss program (12%) on fat oxidation and the colocalization of the fatty acid translocase FAT/CD36 with CPTI in human skeletal muscle.	Caloric intake 500–800kcal/day below that required to maintain body weight. 45 min 3 d/wk of stationary bicycle at 70–85% of HRmax.	FAT/CD36 and CPT1 proteins coimmunoprecipitate in skeletal muscle. Weight loss program (diet + exercise) increased the coimmunoprecipitate of these proteins (25%) and total fat oxidation (R^2^ = 0.857).
Talanian et al., 2007 [34]	Prospective study.Eight healthy recreationally active women: age = 22.1 ± 0.2 y, BM = 65 ± 2.2 kg.	To examine the effects of seven HIIT sessions over 2 wk on skeletal muscle fuel content, mitochondrial enzyme activities, fatty acid transport proteins, VO_2_ peak, and whole body metabolic, hormonal, and cardiovascular responses	Seven HIIT supervised sessions in 13 days: ten 4-min cycling bouts at 90% of VO_2_ peak separated by 2 min of rest.	There were increased whole body fat oxidation and muscle skeletal FABP4, whereas FAT/CD36 content was unaffected.
Talanian et al., 2010 [35]	Prospective study.Ten healthy females: age= 22 ± 1 y, BM = 65 ± 2 kg, VO_2_ peak = 2.82 ± 0.14 L/min.	To determine whether HIIT increased total skeletal muscle, sarcolemmal, and mitochondrial FABP contents.	Three days/wk, completing 18 supervised training sessions in 6 wk: ten 4-min cycling bouts at 90% VO_2_ peak separated by 2 min of rest.	HIIT (3 d/wk for 6 wk) increased the mitochondrial (51%) and whole skeletal muscle amount (10%) of FAT/CD36 without alterations in sarcolemmal content. Whole muscle FABP4 increased in 48%. Sarcolemmal FABP4 increased in 23%, whereas mitochondrial FABP4 was unaltered.
Tunstall et al., 2002 [36]	Cross-sectional and prospective study.Seven heathy untrained individuals: 3 male and 4 female, age = 20–42 y, BMI = 17–26 kg/m^2^	To study the effects of a single bout of exercise and exercise training on the expression of genes necessary for the transport and β-oxidation of FAs, together with the gene expression of transcription factors implicated in the regulation of FA homeostasis.	9-day of 60 min cycling per day at 63% VO_2_ peak.	Nine consecutive days of aerobic training (63% of VO_2_ peak) increased total lipid oxidation and expression of FAT/CD36 and CPT I mRNA during 1-h cycling bout in skeletal muscle.
Warren et al., 2020 [37]	Prospective study.Fourteen premenopausal women: age = 31.2 ± 6.7 y, BMI = 26.6 ± 5.1 kg/m^2^	To assess the effects of aerobic exercise training on skeletal muscle mitochondrial function and markers of lipid metabolism.	Three days per week for 8–16 weeks: 20–40 min, 67–80% of maximal HR.	8–16 weeks of aerobic exercise training increase skeletal muscle FAT/CD36 content and mitochondrial respiratory capacity proportionally.

β-HAD = β-hydroxy acyl-CoA dehydrogenase, AMPK = 5-AMP-activated protein kinase, BM= Body mass, BMI = Body Mass Index, CHO = Carbohydrate, CoA = Coenzyme A, CPT = Muscle Carnitine Palmitoyl Transferase, CPT1 = Carnitine Palmitoyl Transferase 1, DHA = Docosahexaenoic acid, EPA= Eicosapentaenoic acid, FABP4 = Muscle Fatty Acid Binding Protein, FAT/CD36 = Fatty Acid Translocase/Cluster of Differentiation 36, FATP = Long-chain Fatty Acid Transport Protein, FAs = Fatty acids, FHD-2 = Family history of diabetes type 2, GI = Glycemic Index, GLUT4 = Glucose transporter 4, HCD = High carbohydrate diet, HIIT= High intensity interval training, HFD = High fat diet, HGI = High Glycemic Index, HR= Heart rate, LCD = Low carbohydrate diet, LFD = Low fat diet, LGI = Low Glycemic Index, LPL = Lipoprotein Lipase, NA = No accredit, PDK = Pyruvate Dehydrogenase Kinase, PFO = Peak Fat Oxidation, PGC= Peroxisome proliferator-activated receptor gamma coactivator, PPAR = Peroxisome proliferator-activated receptors, RMR = Resting metabolic rate, VO_2_ max = Maximal oxygen uptake, UCP = Uncoupled Protein, VT = Ventilatory threshold. For more information, see the list of abbreviations.

**Table 2 jcm-12-00318-t002:** Post-translational modifications in FAT/CD36.

Post-Translational Modifications	Impact
Glycosylation	Correct folding, stability, transport to the cell surface, and function of the protein.
Acetylation	The consequences for FAT/CD36 expression and/or functioning have not yet been investigated.
Phosphorylation	This process is linked to FAT/CD36 functioning and regulation of FA utilization in the heart and muscle.
Ubiquitination	Considered to trigger the degradation of FAT/CD36 proteins by directing them to proteasomes in skeletal muscle and adipocyte.
Palmitoylation	Have an impact on the subcellular localization, membrane interactions, and subcellular trafficking of proteins.

## Data Availability

The data is available at: https://www.dropbox.com/sh/taf4k3tasuxzrzv/AAAdfgyFE1beP3In5U-t7_23a?dl=0 (accessed on 15 April 2022).

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
