# Peer review of "FAT/CD36 Participation in Human Skeletal Muscle Lipid Metabolism: A Systematic Review"

_jcm, 2022, doi:10.3390/jcm12010318_

Round 1
Reviewer 1 Report
This review presents an interesting topic, particularly in the mechanisms of lipid metabolism in skeletal muscle. However, there is a lack of rigor in the writing of the manuscript:
Introduction
· I suggest that the introduction requires better cohesion and development of ideas/approaches. The context of the problem giving rise to the review must be clearly presented (i.e., including the basic concepts, e.g., CD36 – FAT/CD36 [See next comments of the results]).
Methods
· I suggest that the methodology must be adequately presented, particularly the Boolean search strategy (i.e., uses descriptors with the “AND” “OR” or “NOT”). The idea is that the search is (to some extent) reproducible or verifiable. Some keywords are not clear, e.g., nutri*.
Results
· The results table should be presented with more rigor, some inaccuracies are presented, for example:
o Estudio de Baker et al., 2015: “…obese (n=6; BMI≤30 kg/m2) sedentary”….obese??. Also with Holloway et al., 2007, 9 obese (BMI≤30 kg/m2, 45±3 y) nondiabetic women.
o Better use of abbreviations/acronyms, in a unified way. What is mCD36 (muscle, mitochondrial or mouse)? In many studies mCD36 is referred to as “mouse CD36” (Doi: 10.1371/journal.pone.0210704; Doi:10.1152/ajpgi.00160.2012; Doi: 10.1016/S0022-2275(20)31573-X). In this sense, it is important to unify the terms throughout the manuscrip, in the case of proteins according to databases such as Uniprot or according to the literature (FAT/CD36, for CD36 in skeletal muscle).
o Unify the order of the information of the participants, sometimes it appears. Some appear as “age then BMI”, others “BMI then age”. I reiterate, consistency and uniformity are important in this type of manuscript.
o Perry et al., 2008, BMI unit of measure is missing. Likewise, in “HIIT (18 h at 90% of VO2peak)”, specify that it is the total of the sessions (although “3 d/wk” is enough)
o Similarly, abbreviations that are well-known are redacting better (e.g., VO2max [instead of VO2max]). Likewise, I suggest using the term "sex" instead of "gender",
o In Holloway et al., 2007, I think this study is missing in the “references”: “Skeletal muscle mitochondrial FAT/CD36 content and palmitate oxidation are not decreased in obese women”
o On the other hand, I suggest adding the FAT/CD36 measurement or evaluation technique. This has informative value in basic biomedical sciences.
Discussion
· In line 149-150: “these three criteria biased the results, much less that it detracts from the studies quality”. Good point, I would also argue that the techniques in these studies are aimed at assessing biological mechanisms (e.g., gene expression, cell signaling...), and what this implies...
· Line 153: CD63?
· Line 176 – 177: What do you mean by the expression “making CD36 one of the main ones”
· Figure 4. The structure is from the PDB entry: 5XBM (X-ray Diffraction, 3.50â„«. “Structure of SCARB2-JL2 complex)? If so, I suggest specifying it, and highlighting that the visualization of the structure was done with SWISS-MODEL.
· Line 255: Confirm if Su & Abumrad's review refers to VAMP2. Perhaps the work of Karunakaran et al. (doi: 10.3390/cells10071833). Please be more specific on this argument.
· Figure 5 requires improving the resolution (e.g., a scalable vector graphics would be a good option).
· Line 260: “oxidation of AG”?
· Line 270: …“triacylglycerols synthesis, and lipid droplet formation”. In skeletal muscle, this increase in TG synthesis would be mainly due to what (e.g., activity of Diacylglycerol acyltransferase - DGAT, channeling of fatty acids and other intermediates...)?. The idea is to be more specific and expand/propose the possible mechanisms
· Line 277 – 278: refer to Table 2
· Line 296: check, “Lipids are the principal font of energy at rest”
· Line 349: “aerobic type”, a very vague term. I suggest using more concrete terms in relation to the load or type of physical effort (doi: 10.1186/s40798-015-0012-1)
· Line 354 – 355: Critically verify whether biogenesis/mitochondrial content is evaluated in these studies (correspondence with the technique used and the “outcome”).
Conclusions
Line 466 – 467: “In normal conditions, low and moderate physical activity increases mitochondrial LCFAs accumulation and oxidation to produce ATP” What do you mean by “mitochondrial accumulation LCFA” (is this related to myosteatosis)?
Final comments:
A greater critical analysis of the review of the literature or of the studies presented is suggested, as well as proposing hypotheses, and new perspectives of studies (it is not only a matter of summarizing the results of the studies).
Author Response
This review presents an interesting topic, particularly in the mechanisms of lipid metabolism in skeletal muscle. However, there is a lack of rigor in the writing of the manuscript:
Introduction
- I suggest that the introduction requires better cohesion and development of ideas/approaches. The context of the problem giving rise to the review must be clearly presented (i.e., including the basic concepts, e.g., CD36 – FAT/CD36 [See next comments of the results]).
R: Thank you, it was corrected
Methods
- I suggest that the methodology must be adequately presented, particularly the Boolean search strategy (i.e., uses descriptors with the “AND” “OR” or “NOT”). The idea is that the search is (to some extent) reproducible or verifiable. Some keywords are not clear, e.g., nutri*.
R: Thank you, it was added
Results
- The results table should be presented with more rigor, some inaccuracies are presented, for example:
- Estudio de Baker et al., 2015: “…obese (n=6; BMI≤30 kg/m2) sedentary”….obese??. Also with Holloway et al., 2007, 9 obese (BMI≤30 kg/m2, 45±3 y) nondiabetic women.
Thank you: The Table 1 was corrected, and its format standardized.
- Better use of abbreviations/acronyms, in a unified way. What is mCD36 (muscle, mitochondrial or mouse)? In many studies mCD36 is referred to as “mouse CD36” (Doi: 10.1371/journal.pone.0210704; Doi:10.1152/ajpgi.00160.2012; Doi: 10.1016/S0022-2275(20)31573-X). In this sense, it is important to unify the terms throughout the manuscrip, in the case of proteins according to databases such as Uniprot or according to the literature (FAT/CD36, for CD36 in skeletal muscle).
Thank you: The Table 1 was corrected, and its format standardized.
- Unify the order of the information of the participants, sometimes it appears. Some appear as “age then BMI”, others “BMI then age”. I reiterate, consistency and uniformity are important in this type of manuscript.
Thank you: The Table 1 was corrected, and its format standardized.
- Perry et al., 2008, BMI unit of measure is missing. Likewise, in “HIIT (18 h at 90% of VO2peak)”, specify that it is the total of the sessions (although “3 d/wk” is enough)
Thank you: The number sessions were added.
- Similarly, abbreviations that are well-known are redacting better (e.g., VO2max [instead of VO2max]). Likewise, I suggest using the term "sex" instead of "gender",
Thank you: The sex was corrected, and its format standardized.
- In Holloway et al., 2007, I think this study is missing in the “references”: “Skeletal muscle mitochondrial FAT/CD36 content and palmitate oxidation are not decreased in obese women”
Thank you: It is the reference 28.
- On the other hand, I suggest adding the FAT/CD36 measurement or evaluation technique. This has informative value in basic biomedical sciences.
Thank you: The suggest was added.
Discussion
- In line 149-150: “these three criteria biased the results, much less that it detracts from the studies quality”. Good point, I would also argue that the techniques in these studies are aimed at assessing biological mechanisms (e.g., gene expression, cell signaling...), and what this implies...
Thank you: It was added
Line 153: CD63?
Thank you: It was corrected
- Line 176 – 177: What do you mean by the expression “making CD36 one of the main ones”
Thank you: The paragraph was corrected.
- Figure 4. The structure is from the PDB entry: 5XBM (X-ray Diffraction, 3.50â„«. “Structure of SCARB2-JL2 complex)? If so, I suggest specifying it, and highlighting that the visualization of the structure was done with SWISS-MODEL.
Thank you. The data was corrected
- Line 255: Confirm if Su & Abumrad's review refers to VAMP2. Perhaps the work of Karunakaran et al. (doi: 10.3390/cells10071833). Please be more specific on this argument.
Thank you: It was added
- Figure 5 requires improving the resolution (e.g., a scalable vector graphics would be a good option).
Thank you: The resolution was improved.
- Line 260: “oxidation of AG”?
Thank you: It was corrected.
- Line 270: …“triacylglycerols synthesis, and lipid droplet formation”. In skeletal muscle, this increase in TG synthesis would be mainly due to what (e.g., activity of Diacylglycerol acyltransferase - DGAT, channeling of fatty acids and other intermediates...)?. The idea is to be more specific and expand/propose the possible mechanisms
Dear reviewer, the idea was modified to focus on the CD36 function.
- Line 277 – 278: refer to Table 2
Thank you. It was corrected
- Line 296: check, “Lipids are the principal font of energy at rest”
Thank you. It was eliminated
- Line 349: “aerobic type”, a very vague term. I suggest using more concrete terms in relation to the load or type of physical effort (doi: 10.1186/s40798-015-0012-1)
Thank you. It was added
- Line 354 – 355: Critically verify whether biogenesis/mitochondrial content is evaluated in these studies (correspondence with the technique used and the “outcome”).
Thank you. You are right, the biogenesis mitochondrial was indirectly measure. The data were corrected.
Conclusions
Line 466 – 467: “In normal conditions, low and moderate physical activity increases mitochondrial LCFAs accumulation and oxidation to produce ATP” What do you mean by “mitochondrial accumulation LCFA” (is this related to myosteatosis)?
Thank you. It was corrected
Final comments:
A greater critical analysis of the review of the literature or of the studies presented is suggested, as well as proposing hypotheses and new perspectives of studies (it is not only a matter of summarizing the results of the studies).
Thank you for the prompt and appropriate recommendations. His suggestions were heeded.

Reviewer 2 Report
This review study illustrated the roles of CD36 in the metabolism of skeletal muscle under healthy and unhealthy conditions, which is very meaningful and comprehensive. However, my major concern is that the mechanisms related to CD36 in FAs oxidation and trafficking in skeletal muscle seem not be discussed thoroughly, therefore, the conclusions and goals did not match very well. More discussion regarding the mechanisms behind the phenomenon is needed.
In addition, here are some minor comments.
Line 11: “among other factors” is redundant. Please either delete it or specify the exact factors.
Line 12: “other cells”- Please specify the cell types.
Line 14: “in some diseases”- please specify the diseases.
Line 31: " among other diseases”- please specify the diseases.
Line 465: “from 464 the interstitium, the cytoplasm, and into the mitochondria” is not grammarly correct.
Line 468: “among other molecules” is redundant.
Author Response
This review study illustrated the roles of CD36 in the metabolism of skeletal muscle under healthy and unhealthy conditions, which is very meaningful and comprehensive. However, my major concern is that the mechanisms related to CD36 in FAs oxidation and trafficking in skeletal muscle seem not be discussed thoroughly, therefore, the conclusions and goals did not match very well. More discussion regarding the mechanisms behind the phenomenon is needed.
Thank you. Considering the suggestions, the manuscript was revised and corrected.
In addition, here are some minor comments.
Line 11: “among other factors” is redundant. Please either delete it or specify the exact factors.
Thank you. It was eliminated
Line 12: “other cells”- Please specify the cell types.
Thank you. The abstract was corrected.
Line 14: “in some diseases”- please specify the diseases.
Thank you. The abstract was corrected.
Line 31: " among other diseases”- please specify the diseases.
Thank you. It was eliminated
Line 465: “from 464 the interstitium, the cytoplasm, and into the mitochondria” is not grammarly correct.
Thank you. It was corrected
Line 468: “among other molecules” is redundant.
Thank you. It was corrected

Reviewer 3 Report
This is a well-designed study with the application of relevant methodology. I have the following comments;
1. It was hard to comprehend the actual sense throughout the document as English is not up to the mark. For example, in Line 11, the sentence begins with 'His' which didn't make any clear sense. I would like to reread the rewritten version to have better and more relevant suggestions/comments.
2. Line 33-Obesity is not a disease in itself but a condition that can lead to multiple adverse conditions.
3. Methods-Literature search strategy- 'Human skeletal muscle' or just 'skeletal muscle' hasn't been used in search terms? In addition 'fat metabolism' is also missing. The title refers to both fat metabolism and human skeletal muscle.
4. Outcomes-Line 101- Did you consider observing the effects of only exercise or diet, or are the effects of both exercise and diet included?
5. Line 115- Which figure is 1? Are you referring to the methodology overview?
6. Table 1 (Bergouignan et al., 2012 and Baker et al., 2015)- Does this mean that increasing the percentage of daily energy intake within a diet can modulate lean individuals' fat metabolism? With 35% it didn't affect but with 50% CD36 and other marker levels increased.
Strange that studies from 2014 to 2019 haven't been included as multiple studies that could fall in the criteria have been performed in this duration.
7. When referring to FABP, which one do authors check for? FABP3 is adipocyte-specific while FABP4 is involved in skeletal muscle.
Thank you.
Author Response
This is a well-designed study with the application of relevant methodology. I have the following comments;
Dear reviewer thank you very much for the questions and suggestions provided.
1. It was hard to comprehend the actual sense throughout the document as English is not up to the mark. For example, in Line 11, the sentence begins with 'His' which didn't make any clear sense. I would like to reread the rewritten version to have better and more relevant suggestions/comments.
Dear reviewer. Thanks for his suggestion; the manuscript was again reviewed, correcting the syntax and prepositions.
- Line 33-Obesity is not a disease in itself but a condition that can lead to multiple adverse conditions.
Dear reviewer, due to the high world prevalence of obesity and been it a systemic inflammatory chronic problem, we consider it a disease. However, his suggestion was respected and eliminated the term disease.
- Methods-Literature search strategy- 'Human skeletal muscle' or just 'skeletal muscle' hasn't been used in search terms? In addition 'fat metabolism' is also missing. The title refers to both fat metabolism and human skeletal muscle.
Dear reviewer, with the methodology used for the search for manuscripts (keywords, tickets, search for manuscripts within the references of the chosen manuscripts), we include all possible articles on the subject. The inclusion criteria also helped us. To confirm what we mentioned here, we performed a new search with the keywords you suggested and found 14 results already in the original database. None of these 14 manuscripts compare with the inclusion criteria.
- Outcomes-Line 101- Did you consider observing the effects of only exercise or diet, or are the effects of both exercise and diet included?
Dear reviewer. As can be seen in Table 1 of results we consider both treatments, together and separated.
- Line 115- Which figure is 1? Are you referring to the methodology overview?
Thanks. The bottom figure was added.
- Table 1 (Bergouignan et al., 2012 and Baker et al., 2015)- Does this mean that increasing the percentage of daily energy intake within a diet can modulate lean individuals' fat metabolism? With 35% it didn't affect but with 50% CD36 and other marker levels increased.
Rear reviewer: According to the studies reviewed an acute and chronic high-fat diet (50-70% of lipids) and/or endurance exercise (<80% of HR) increase the human skeletal muscle FAT/CD36 mRNA and protein content; its translocation to the cell membrane and mitochondria increases lipid oxidation.
Strange that studies from 2014 to 2019 haven't been included as multiple studies that could fall in the criteria have been performed in this duration.
Dear reviewer. The publication date was not a search or selection criterion.
- When referring to FABP, which one do authors check for? FABP3 is adipocyte-specific while FABP4 is involved in skeletal muscle.
Thank you. It was corrected

Round 2
Reviewer 1 Report
The work of the authors to improve the manuscript is highlighted, according to the comments of the review. There is a better order in the content and writing, in general the text is more organized.
I congratulate the authors for the work done.